# Ovarian Cancer in the Elderly: Time to Move towards a More Logical Approach to Improve Prognosis—A Study from the FRANCOGYN Group

**DOI:** 10.3390/jcm9051339

**Published:** 2020-05-04

**Authors:** Ludivine Dion, Camille Mimoun, Krystel Nyangoh Timoh, Sofiane Bendifallah, Alexandre Bricou, Pierre Collinet, Cyril Touboul, Lobna Ouldamer, Henri Azaïs, Yohann Dabi, Cherif Akladios, Geoffroy Canlorbe, Pierre-Adrien Bolze, Hélène Costaz, Mathieu Mezzadri, Tristan Gauthier, Frederik Kridelka, Pauline Chauvet, Nicolas Bourdel, Martin Koskas, Xavier Carcopino, Emilie Raimond, Olivier Graesslin, Lise Lecointre, Marcos Ballester, Jean Levêque, Cyrille Huchon, Vincent Lavoué

**Affiliations:** 1Department of Gynecology, Rennes University Hospital, 35000 Rennes, France; ludivine.dion@chu-rennes.fr (L.D.); krystel.nyangoh.timoh@chu-rennes.fr (K.N.T.); jean.leveque@chu-rennes.fr (J.L.); 2IRSET, Equipe 8, INSERM University 1085, 35000 Rennes, France; 3Department of Gynecology and Obstetrics, Lariboisiere Hospital, 750019 Paris, France; camille.mimoun@aphp.fr (C.M.); matthieu.mezzadri@aphp.fr (M.M.); cyrillehuchon@yahoo.fr (C.H.); 4Service de Gynécologie, Hopital TENON, AP-HP, 75020 Paris, France; sofiane.bendifallah@aphp.fr (S.B.); cyril.touboul@aphp.fr (C.T.); 5Service de Chirurgie Gynécologique, Hôpital Croix Saint Simon, 75000 Paris, France; alex.bricou@gmail.com (A.B.); mballester@hopital-dcss.org (M.B.); 6Service de Gynécologie, Hôpital Jeanne de Flandres, 59000 Lille, France; Pierre.COLLINET@CHRU-LILLE.FR; 7Service de Gynécologie, 37000 Tours, France; louldamer@yahoo.fr; 8Service de Gynécologie, Hôpital La Pitié Salpétrière, 75012 Paris, France; henri.azais@aphp.fr (H.A.); geoffroy.canlorbe@aphp.fr (G.C.); 9Service de Gynécologie Obstétrique, Centre Hospitalier Inter Communal de Créteil, 94000 Créteil, France; yohann.dabi@aphp.fr; 10Service de Gynécologie Obstétrique, CHU Hautepierre, 67000 Strasbourg, France; cherif.akladios@gmail.com (C.A.); lise.lecointre@chru-strasbourg.fr (L.L.); 11Service de Gynécologie Obstétrique, CHU Lyon Sud, 69000 Lyon, France; pierre-adrien.bolze@chu-lyon.fr; 12Département d’Oncologie Chirurgicale, Centre Georges François Leclerc, 21000 Dijon, France; hcostaz@cgfl.fr; 13Service de Gynécologie Obstétrique, CHU de Limoges, 87000 Limoges, France; Tristan.Gauthier@chu-limoges.fr; 14Service de Chirurgie Oncologique et Gynécologique, 4000 Liège, Belgique; frederic.kridelka@chu.ulg.ac.be; 15Service de Gynécologie Obstétrique, CHU de Clermont Ferrand, 63000 Clermont Ferrand, France; po.chauvet@gmail.com (P.C.); nicolas.bourdel@gmail.com (N.B.); 16Service de Gynécologie, Hôpital Bichat, APHP, 75018 Paris, France; martin.koskas@aphp.fr; 17Service de Gynécologie, Hôpital La Timone, APHM, 13000 Marseille, France; xcarco@free.fr; 18Service de Gynécologie, Hôpital Universitaire de Reims, 51000 Reims, France; Emilie.raimons@chu-reims.fr (E.R.); olivier.graesslin@gmail.com (O.G.)

**Keywords:** ovarian cancer, elderly, surgery, chemotherapy, frailty

## Abstract

Background and objective: Elderly and/or frail women with ovarian cancer are often undertreated. The aim of the study is to compare the effects of age and frailty on surgical approaches, postoperative complications, and prognosis in elderly women with ovarian cancer. Methods: A retrospective multicenter study of women ≥70 years were treated for ovarian cancer at seven French university hospitals between 2007 and 2015. Results: Of the 1119 women treated for ovarian cancer during the study period, 147 were ≥70 years and had complete data. Of these women, 65 were aged 70–74 years, and 82 were aged ≥75 years. Overall, 77% of the younger women (49/65) received optimal treatment compared with 51% (40/82) of the older women (*p* = 0.018). Women ≥75 years underwent fewer bowel resections (32% vs. 67%, *p* < 0.001) and experienced fewer postoperative complications (22.6% vs. 38.9%, *p* < 0.001); 53.2% of the women in this age group were treated by primary surgery or surgery only. These women also received more chemotherapy with platinum only (15% [9/56] vs. 2% [1/57], *p* = 0.007) and less bevacizumab (9% [5/56] vs. 32% [18/57], *p* = 0.003). Patients with greater frailty (a modified Charlson Comorbidity Index [mCCI] score >3) had a five-year survival rate of 30% versus 62% for those with a score ≤3 (*p* < 0.001). Conclusions: Surgeons modify their approach to treating ovarian cancer in women ≥75 years probably to reduce immediate postoperative complications. The prognosis is significantly worse in patients with greater frailty. Improvements to the sequence of treatments administered, with priority given to neoadjuvant chemotherapy in patients with greater frailty, could help increase the number of women who receive optimal treatment and improve their prognosis.

## 1. Introduction

With an increasing life expectancy in the Western world, the incidence of cancer in the elderly is on the rise: an estimated 50% of all cancer cases is diagnosed in patients aged >65 years [1]. In France, the number (and proportion) of people aged >75 years increased from 1.5 million (3.8% of the total population) in 1950 to 6 million (9.3%) in 2018, and this figure is expected to reach 12 million (16%) by 2050 [2]. The average age at diagnosis of ovarian cancer is 68 years [3]. Despite the existence of new targeted therapies such as bevacizumab and olaparib, prognosis in ovarian cancer remains poor, with an overall five-year survival rate of 43% [3]. It is even worse in women aged >75 years, who have a five-year survival rate of just 25% [1]. These high mortality rates can largely be explained by the presence of comorbid conditions and frailty in this population, as these factors impede optimal management [3]. Several studies have shown that elderly women do not receive the standard recommended treatment for ovarian cancer. In short, they are less likely to undergo complete cytoreductive surgery or be treated with a full chemotherapy regimen (in terms of doses, number of cycles, and drug components) [4,5,6]. One study found that just 20% of women with ovarian cancer aged >80 years received optimal treatment [7].

Decisions to adjust treatment protocols are often guided by chronological age: the importance attached to age is evident in clinical trials which tend to recruit young patients [5,8]. Chronological age alone, however, should not be used to guide treatment decisions, as it does not take frailty into account [9]. Several studies have shown that frailty scores perform better than chronological age at predicting prognosis [10]. Surgical insult is a form of acute stress that requires an increased physiological response from frail patients at risk of postoperative complications. Surgeons often choose to simplify surgery in the hope of reducing complications and improving survival, but this decision is often based on chronological age alone. Frailty indices such as the modified Charlson Comorbidity Index (CCI), which has a strong predictive accuracy for mortality, could be useful for guiding treatment decisions in such cases [5,11,12]. Several studies have shown that the use of modified treatment regimens in elderly patients could reduce postoperative complications without altering long-term survival [4,12,13]. The modified Charlson Comorbidity Index (mCCI) corresponds to the age adjusted CCI.

The aim of the study is to compare the effects of age and frailty—assessed using the modified CCI (mCCI)—on surgical approaches, postoperative complications, and prognosis in elderly women with ovarian cancer.

## 2. Material and Methods

### 2.1. Design

We conducted a retrospective, multicenter, observational study in which we collected data on patients aged >70 years treated for ovarian cancer at seven French university hospitals (CHU de Tenon, Jean Verdier, Lille, Rennes, Poissy, Tours, Créteil, and Strasbourg) between 2007 and 2015. The participating hospitals each perform over 20 ovarian cancer operations a year.

The study was approved by the ethics committee of the National College of French Gynecologists and Obstetricians (CEROG 2016-GYN-1003) and patients were duly informed about the study, as required by French law.

### 2.2. Management of Patients with Ovarian Cancer

Treatments were administered according to the sequence recommended by the French Clinical Practice Guidelines on the Management of Epithelial Cancer of the Ovary, Fallopian Tube, and Primary Peritoneum [14]. In brief, women diagnosed with ovarian cancer underwent a staging laparoscopy immediately followed by laparotomy debulking surgery when a complete resection of all macroscopically visible disease was deemed possible. Otherwise, they were treated with neoadjuvant chemotherapy followed by interval debulking surgery and at least six cycles of adjuvant chemotherapy. The standard surgical staging procedures consisted of hysterectomy, bilateral adnexectomy, omentectomy, appendectomy, pelvic and para-aortic lymph node dissection (inclusion period before the LION (lymphadenectomy in ovarian neoplasms) clinical trial [15]), and any additional procedures considered necessary to achieve complete macroscopic clearance. Modifications to the treatment protocol were made at the discretion of the medical and surgical team.

### 2.3. Data Collection

The following variables were collected: age at diagnosis, body mass index (BMI) assessed as kg/m^2^, parity, presence of comorbidities (high blood pressure, diabetes, chronic neurological disease, chronic lung disease, dyslipidemia, cardiovascular disease, autoimmune disease), and data on genetic predisposition. Weight loss was defined by a loss of weight during the year before the diagnostic of the ovarian cancer (i.e., loss of at least one kilogram).

The mCCI score was calculated for each patient [12,16]: the Charlson Comorbidity Index is adjusted by the following items: One point, if there is history of myocardial infarction, congestive heart failure, peripheral vascular disease, cerebrovascular accident, dementia, chronic obstructive pulmonary disease, connective tissue disease, peptic ulcer disease, liver disease without portal hypertension, or diabetes mellitus uncomplicated. Two points, if there is history of hemiplegia, moderate to severe chronic kidney disease, diabetes mellitus with organ damage, a solid tumor, leukemia, or lymphoma. Three points, if there is history of liver disease with portal hypertension. Six points if there is a history of a metastatic solid tumor or acquired immune deficiency syndrome. The mCCI score corresponds to the CCI score adjusted by the age, one point is added between 50 and 59 years, two points between 60 and 69 years, three points between 70 and 79 years, four points between 80 and 89 years, and five points after 89 years.

The surgical data collected included the Fagotti laparoscopic score [17], the Peritoneal Cancer Index (PCI) score (tumor burden during debulking laparotomy) [18], the sequence of treatments (primary surgery vs. interval debulking surgery), surgical procedures performed, and presence or absence of residual disease at the end of the intervention. Surgical complexity was classified using the model described by Aletti et al. [19]. The following histologic data were collected: histologic subtype (serous, mucinous, endometrioid, clear-cell, or carcinosarcoma), stage, and lymph node involvement. Disease was staged using the FIGO (International Federation of Gynecology and Obstetrics) system [20], and a note was made of the number of chemotherapy cycles administered.

Postoperative complications (occurring within 30 days of surgery) were classified according to the Clavien–Dindo system [21] as minor for grade I and II complications and major for grade III, IV, and V complications. A note was also made of the date of the last follow-up, disease recurrence (defined as an increase in CA 125 levels or evidence of recurrence by imaging and/or histology), death, and date of death.

### 2.4. Outcome Measures

The main outcome measure was the complete resection rate, i.e., proportion of patients without macroscopically visible disease. Secondary measures were postoperative complications, tumor recurrence, and death.

### 2.5. Statistical Analyses

The women were divided into four groups for comparison: women aged 70–74 versus ≥75 years and women with an mCCI score of ≤3 versus >3.

Age cut-off was chosen according to the literature [22] and but also because after 75 years, a geriatric check-up is necessary before beginning treatment. The mCCI score cut-off was chosen according to the first publication of the Charlson Comorbidity Index [16]: authors showed that a high mortality rate is correlated with a cut-off of 2 or 3.

Descriptive parameters are expressed as the median (range) or percentages, as appropriate. Demographic and clinical (tumor and treatment) characteristics were compared according to age and mCCI scores using the χ^2^ for categorical variables and the *t* test for continuous variables. Overall survival was defined as the time from surgery to death (of any cause) or date of last follow-up. Disease-free survival was defined as the time from surgery to cancer recurrence. Women who remained alive and whose disease did not progress were censored at the date of the last follow-up. The Kaplan–Meier method was used to build survival curves, which were compared using the log-rank test. A multivariate analysis was conducted to identify the predictors of postoperative complications and the presence of residual disease. The final models were obtained by backward stepwise selection of the variables from the full model with a retention threshold of *p* < 0.20; the hierarchy principle for retaining variables was applied. In the multivariate analysis, missing data were considered as absent. A p-value of <0.05 was considered statistically significant.

## 3. Results

Of the 1119 women treated for ovarian cancer at one of the seven participating hospitals between 2007 and 2015, 269 were ≥70 years. Of these, 122 patients were excluded because of lacking data that could not allow a CCI score to be performed. Thus, 147 of these had complete study data and were included. Sixty-five women were aged 70–74 years and 82 were ≥75 years.

### 3.1. Demographic and Clinical (Tumor and Treatment) Characteristics

Demographic and tumor characteristics stratified by age are shown in Table A1. Treatment characteristics stratified by age and mCCI score are shown in Table A2 and Table A3, respectively. Demographic and tumor characteristics were similar in the two mCCI groups (data not shown) except for age. Women with an mCCI ≤3 were significantly younger than those with an mCCI >3 (median age, 73 years [70–79 years] vs. 79.5 years [72–99 years]; *p* < 0.001). The BMI was higher in the 70–74-year group than in the ≥75-year group and was, respectively, 25.8 and 23.0 kg/m^2^ (*p* = 0.006). The number of patients’ weight loss was significantly more important in the 70–74-year group than in the ≥75-year group and was, respectively, 12.3% and 23.2% (*p* = 0.041). Concerning the CCI score, there was no significant difference in the two age groups and was 0.43 (0–4) for the younger and 0.39 (0–5) for the older (*p* = 0.529) group. Concerning the tumor characteristics, no significant difference was shown between the two age groups: serous tumors represented the more histological tumor and were 70.8% (46/65) for the younger and 70.7% (58/82) for the older (*p* = 0.862) group. There was no significant difference regarding the grade (high or low) of the serous tumor. The FIGO stage was similar between the two age groups.

### 3.2. Predictors of Postoperative Complications and Complete Resection

No associations were observed between perioperative complications and age, BMI, histologic subtype, FIGO stage, treatment sequence, or lymph node involvement (data not shown) either in the univariate or multivariate analyses (Table A4). The corresponding results for complete resection are shown in Table A5.

#### Follow-up

The median recurrence-free survival was 23 months (1–150 months), and the five-year recurrence-free survival rate was 25%. The median overall survival was 31 months (1–150 months), and the five-year survival rate was 20%.

Overall and recurrence-free survival curves according to age and mCCI score are shown in Figure 1.

Median recurrence-free survival was 144 months (1–88 months) for patients between 70 and 74 years versus 76 months (1–151 months) for patients older than 75 years; the respective five-year survival rates were 59% and 61% (*p* = 0.481). The median overall survival was 45 months (1–150 months) for patients between 70 and 74 years versus 35 months (1–123 months) for patients older than 75 years, but the five-year survival rate was identical, at 32.5%.

The median recurrence-free survival was 46 months for patients with an mCCI ≤3 versus 76 months for those with an mCCI >3 (*p* = 0.243). The respective five-year recurrence-free survival rates were 37% and 53%. The median overall survival was 114 months in women with an mCCI ≤3 versus 28 months in those in with an mCCI >3 (*p* < 0.001). The corresponding five-year survival rates were 62% and 30%.

## 4. Discussion

In this series of women with ovarian cancer aged ≥70 years, women ≥75 years received different treatment to those aged 70–74 years, despite having similar tumor characteristics. In particular, they received less combined treatment with surgery and chemotherapy (51% vs. 77%, *p* = 0.018), underwent fewer bowel resections (39.5% vs. 55.2%, *p* = 0.020), and were treated less often with bevacizumab (8.9% vs. 31.5%, *p* = 0.003) and more often with a mono-chemotherapy (14.8% vs. 1.8%, *p* = 0.007). On analyzing treatments by frailty, the patients in the two mCCI groups (≤3 vs. >3) had similar results, although surgical complexity was lower in patients with an mCCI score >3, suggesting that surgeons adopt a simpler approach in frailer patients to limit postoperative complications. Indeed, complications were less frequent in women with an mCCI score >3 (23.5%) than in those with an mCCI score ≤3 (35.7%) (*p* = 0.223).

We showed comparable complete resection rates in the four groups (70.4% vs. 66.7% in women aged 70–74 years vs. ≥75 years and 68.7% and 69.7% in women with an mCCI score of ≤3 vs. ≥3), as we would have expected to see more cases of residual tumor in women who underwent less complex surgery. This may indicate that surgical evaluation is influenced by subjective criteria or that it does not include a full examination of the peritoneal cavity in older (≥75 years) and/or frailer women. The question of subjective operative assessment of residual disease at the end of an intervention was reported in other studies, although not specifically linked to age [23,24,25]. Ezkander et al. analyzed 639 women who underwent primary surgery for stage III ovarian cancer and who had been evaluated as having achieved an optimal outcome (residual disease ≤1 cm) by the surgeons [23]. However, postoperative computed tomography (CT) revealed that 40% of the women had a more extensive disease (>1 cm). This discrepancy was accompanied by significantly shorter recurrence-free survival in women with discordant results (median, 12.8 months) than in those in whom the surgeons’ clinical assessment coincided with the CT findings (median, 18.3 months) (*p* = 0.0059). Similar results have been reported by Burger et al. [24] and Chi et al. [25].

In our study, we also observed that the sequence of treatments varied according to age and frailty, although this decision may not have been entirely based on objective criteria. Optimal treatment (combination of surgery and chemotherapy, in whatever order) was considerably less frequent in women ≥75 years (20.3% vs. 40.6% for women aged 70–74 years) and in women with an mCCI score >3 (22.6% vs. 43.6% for those with a score ≤3). In a subanalysis of data from a trial conducted by Vergotte et al. [26], women with a performance status of two (indicating greater frailty) who received neoadjuvant chemotherapy had higher survival rates than those treated with primary surgery. In addition, those who underwent neoadjuvant chemotherapy had higher complete resection rates (81% vs. 42%) and lower postoperative complication rates. It would probably therefore be of interest to reduce primary surgery rates in patients with greater frailty—in our series, 50% of patients with an mCCI score >3 underwent primary surgery (27.4%) or surgery only (22.6%). Reducing primary surgery in these patients would lead to a greater use of interval cytoreductive surgery and reduce the proportion of frail patients treated with surgery only (22.6% in our series vs. 8.6% for those with an mCCI <3). Favoring CT-guided biopsy over primary surgery for histologic examination followed by neoadjuvant chemotherapy and interval cytoreductive surgery in frail patients seems like a wise choice, as it would increase the number of patients treated with a combination of chemotherapy and surgery. The risk of postoperative complications associated with primary surgery precludes chemotherapy in frail patients.

The undertreatment observed in our series probably explains the poor prognosis observed in women with an mCCI score >3 and is consistent with previous reports [27]. The challenge therefore probably lies in improving overall management by prioritizing, for example, neoadjuvant chemotherapy in frail patients, as recommended by the French guidelines [14]. Objective operative assessment of residual disease following debulking also needs to be improved. Although it has been shown that with careful monitoring, bevacizumab does not increase the risk of toxicity in patients >65 years [28,29], in our series, this drug was used significantly less often in older patients (9% of patients ≥75 years vs. 32% of those aged 70–74 years, *p* = 0.003). When used appropriately, bevacizumab can improve overall survival in patients with a worse prognosis, namely those with residual tumor after cytoreductive surgery [30]. A recent French study (EWOC-1 trial) showed that for vulnerable elderly patients with ovarian cancer, carboplatin monotherapy or weekly carboplatin plus paclitaxel are often proposed as an alternative to carboplatin plus paclitaxel given every three weeks but they showed that compared to the three-weekly and weekly carboplatin plus paclitaxel regimens, the carboplatin single agent was reported to be less active with a significantly worse survival outcome in vulnerable elderly patients, in vulnerable patients carboplatine paclitaxel combination remains a standard [31].

This study has some limitations related to its retrospective design, including missing data and potential selection bias. There are, however, no prospective studies on the effects of age or frailty on modifications to surgical approaches, postoperative complications, or prognosis in elderly women with ovarian cancer. Our sample was similar in size to most of the series published to date. Cohort studies and, where possible, controlled randomized trials involving elderly women with ovarian cancer, are urgently needed to identify optimal treatment protocols—assessed using objective criteria—for frail women.

## 5. Conclusions

Chronological age is an insufficient indicator of frailty, but perceived frailty based on subjective criteria in elderly woman with ovarian cancer may lead surgeons to simplify their surgical approach and possibly reduce the thoroughness of their operative assessment of residual disease, which subsequently precludes the objective consideration of the appropriateness of targeted treatments such as bevacizumab. This “intuitive” approach may also result in the suboptimal application of treatment sequences, as primary surgery is too often given priority over neoadjuvant chemotherapy, which would probably allow patients with greater frailty to benefit from combined treatment with chemotherapy and surgery. In conclusion, the objective assessment of frailty using validated scales before treatment would increase the use of optimal treatment sequences in frail women, improving their overall survival.

## Figures and Tables

**Figure 1 jcm-09-01339-f001:**
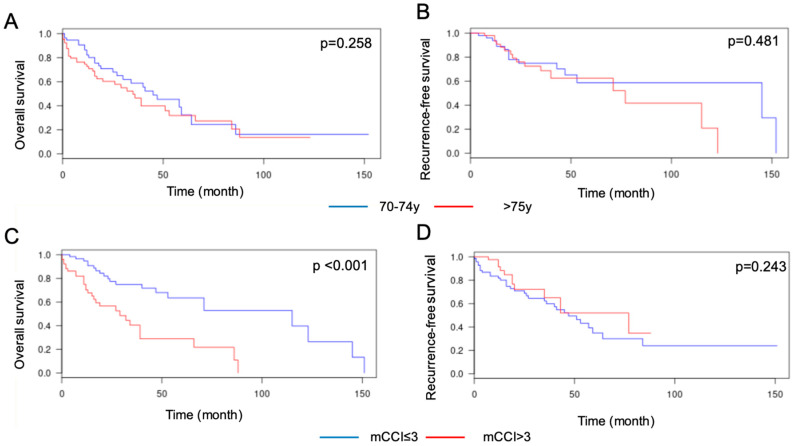
Kaplan–Meier curves estimating overall and recurrence-free survival by age and modified Charlson Comorbidity Index (mCCI) score; (**A**) overall survival by age; (**B**) recurrence-free survival by age; (**C**) overall survival by mCCI score; (**D**) recurrence-free survival by mCCI score.

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
