# Peer review of "Ovarian Cancer in the Elderly: Time to Move towards a More Logical Approach to Improve Prognosis—A Study from the FRANCOGYN Group"

_jcm, 2020, doi:10.3390/jcm9051339_

Round 1
Reviewer 1 Report
The authors have analyzed the effects of age and frailty—assessed using the modified CCI (mCCI)—on surgical approaches, postoperative complications, and prognosis in elderly women with ovarian cancer..
There are several issues that need be addressed to clarify the paper.
Major:
- Aim of the abstract should be rephrased as this is not what the authors analyzed. The aim of the introduction is good.
- The authors describe in their methods that they analyze the mCCI aacroding to ref 12. However, for the paper its is of importance that this is briefly explained and that you don’t have to search for the other paper.
- In their statistical part of the methods the authros describe the following “The women were divided into four groups for comparison: women aged 70–74 versus ≥75 years (Groups 1 and 2, respectively) and women with an mCCI score of ≤3 versus >3 (Groups 3 and 4, respectively).” However, based on which criteria did they decide to use these cut-offs?
In addition, they show in table 1 that the mCCI ranges from 3-7 in the “young” group and 3-9 in the “old” group, so why can you then make a cut-off <3 and >=3?
- The authors mention that they have only complete data from 147 patients out of the 269. What is complete? And couldn’t they include more patients despite they have less data, but for example ~80% is available? In this way they have more power to perform their analyses.
Moreover, why do they only show data of 109 iso 149 patients in table 4?
- In table 1, the authors show weight loss. What is meant by 8? the number of women with weight loss? what is defined as weight loss? more than XX kg? percentagesa re not correct, for both groups.
- In table 1, what is meant by Kg, mean?
- For the whole results section, please describe more, and that not the reader have to extract all data out of the table including drawn conclusions.
- Layout of figure and tables should be improved.
Minor:
- In the introduction the authors describe that “The Charlson Comorbidity Index (mCCI) is the age-adjusted CCI.” However, before the mention the following “Frailty indices such as the modified Charlson Comorbidity Index (CCI),”. Seems now that it is simimalr, but this should not be. Please correct.
- Grade should also be included. Especially the difference between high-grade serous and low-grade This should also be adjusted in the tables.
- In the discussion, the authors mention that they are surprised by their data. This is not very scientific. Please rewrite.
Author Response
The authors want to thank the reviewer for their highlights and very clever comments. Please find below answer to comments.
- Aim of the abstract should be rephrased as this is not what the authors analyzed. The aim of the introduction is good.
The authors agree with this comment and they rephrased the aim of the abstract changing sentence : “The aim of this study was to evaluate the nature of this undertreatment ” by “The aim of the study was to compare the effects of age and frailty on surgical approaches, postoperative complications, and prognosis in elderly women with ovarian cancer.”
- The authors describe in their methods that they analyze the mCCI according to ref 12. However, for the paper its is of importance that this is briefly explained and that you don’t have to search for the other paper.
The authors agree with this comment andthey explained the mCCI score in the materiel and methods section and added the citation of the first describe of the CCI score : the Charlson Comorbidity Index is adjusted by the following items : 1 point if history of myocardial infarction or congestive heart failure or peripheral vascular disease or history of a cerebrovascular accident or dementia or chronic obstructive pulmonary disease or connective tissue disease or peptic ulcer disease or liver disease without portal hypertension or diabetes mellitus uncomplicated. Two points if hemiplegia or moderate to severe chronic kidney disease or diabetes mellitus with organ damage or solid tumor or leukemia or lymphoma. Three points if liver disease with portal hypertension. Six points if metastatic solid tumor or acquired immune deficiency syndrome. The mCCI score corresponds to the CCI score adjusted by the age, 1 point is added between 50-59 years, 2 points between 60-69 years, 3 points between 70-79 years, 4 points between 80-89 years and 5 points after 89 years.
- In their statistical part of the methods the authros describe the following “The women were divided into four groups for comparison: women aged 70–74 versus ≥75 years (Groups 1 and 2, respectively) and women with an mCCI score of ≤3 versus >3 (Groups 3 and 4, respectively).” However, based on which criteria did they decide to use these cut-offs?In addition, they show in table 1 that the mCCI ranges from 3-7 in the “young” group and 3-9 in the “old” group, so why can you then make a cut-off <3 and >=3?
The reviewer is right. Authors agree that the threshold of 75 years of age could be an arbitrary choice, but there is no consensual threshold in literature to defined elderly or very elderly. Thus, age cut-off was chosen because this cut-off of 75 years of age was often chosen and because after 75 years a geriatric check-up is necessary before to begin an oncologic treatment. The mCCI score cut-off of 3 was chosen according to the first publication of the Charlson Comorbidity Index: authors showed that highmortality rate is correlated with a cut-off of 2 or 3.Authors added these information in the materiel and methods section.
4 The authors mention that they have only complete data from 147 patients out of the 269. What is complete? And couldn’t they include more patients despite they have less data, but for example ~80% is available? In this way they have more power to perform their analyses.
The authors agree with this comment. Full data that allowed calculation of CCI score was only available for 149 patients and if we cannot perform CCI score, the design of study could not be performed. A sentence was added in result section for the 122 patients that were excluded.
- Moreover, why do they only show data of 109 iso 149 patients in table 4?
The reviewer is right. We added a sentence in material and methods that indicated : “In the multivariate analysis, missing data were considered as absent”. Indeed, for the multivariate analysis, if a data was missed in the model we have to remove patient, that is why we have to remove some patient. In table 4 we removed the column of n/N because it was the N for the multivariate analysis and it was confused for the univariate analysis in the same table.
- In table 1, the authors show weight loss. What is meant by 8? the number of women with weight loss? what is defined as weight loss? more than XX kg? percentages are not correct, for both groups.
Theauthors agree with this comment: Weight loss was defined by a loss of weight during the year before the diagnostic of the ovarian cancer. The explanation has been added in the materiel and methods section.
Percentages was verified and modified in the table 1. 8 was the number of patient with weight loss. 12,3% in the young group and 23,2% in the old group, result is significative p value is 0,041.
- In table 1, what is meant by Kg, mean?
It was the level of weight loss expressed in kg, meaning kilogram. Table was modified to increase the understanding.
For the whole results section, please describe more, and that not the reader have to extract all data out of the table including drawn conclusions.
The authors agree with this comment and they have added the results of the table in the result section.The BMI was higher in the 70-74y group than in the ≥75y and is respectively 25.8 and 23.0 kg/m2 (p=0.006). The number of patients’ weight loss is significantly more important in the 70-74y group that in the ≥75y and is respectively 12.3% and 23.2% (p=0.041). Concerning the CCI score, there is no significantly difference in the two age groups and was 0.43 (0-4) for the younger and 0.39 (0-5) for the older (p=0,529). Concerning the tumor characteristic, no significantly different was shown between the two age groups: serous tumor represent the more represent histological tumor and was 70.8% (46/65) for the younger and 70.7% (58/82) for the older (p=0.862). There is no significant difference regarding the grade (high or low) of the serous tumor. The FIGO stage was similar between the two age groups.
- Layout of figure and tables should be improved.
Layout of figure and tables was improved.
- In the introduction the authors describe that “The Charlson Comorbidity Index (mCCI) is the age-adjusted CCI.” However, before the mention the following “Frailty indices such as the modified Charlson Comorbidity Index (CCI),”. Seems now that it is similar, but this should not be. Please correct.
The authors agree with this comment andhave modified the sentence : “The Charlson Comorbidity Index (mCCI) is the age-adjusted CCI.” By the “The modified Charlson Comorbidity Index (mCCI) correspond to the age-adjusted CCI”
11 Grade should also be included. Especially the difference between high-grade serous and low-grade This should also be adjusted in the tables.
Table 1 was modified by adding the number of patients with high and low grade serous ovarian cancer as following
|
Histologic subtype Serous Serous high grade Serous low grade Unknow Mucinous Endometrioid Clear-cell Other Undifferentiated Unknow |
46 (70,8) 35 (76,1) 3 (6,5) 8 (17,4) 4 (6.2) 6 (9.2) 2 (3.1) 3 (4.6) 2 (3.1) 2 (3.1)
|
58 (70,7) 38 (65,5) 7 (12,0) 13 (22,4) 3 (3,7) 4 (4,9) 3 (3,7) 6 (7.3) 1 (1.2) 7 (8,5) |
0.862
|
In the discussion, the authors mention that they are surprised by their data. This is not very scientific. Please rewrite.
The authors agree with this comment and changed the sentence by :”We found a comparable complete resection rates in the four groups”

Reviewer 2 Report
Page 3 after laparoscopy patients had laparotomy surgery!!!!
I wonder why the probability of complete resection was much lower in patients with stage less than 2 a (page 6 table 4)
Thee are a number of biases in this study as the Authors suggest. For these reason I don't believe it adds any significant contribution to the literature
Author Response
The authors want to thank the reviewer for their highlights and very clever comments. Please find below answer to comments.
- Page 3 after laparoscopy patients had laparotomy surgery!!!!
The authors agree with this comment and have corrected the word.
The sentence : « In brief, women diagnosed with ovarian cancer underwent a staging laparoscopy immediately followed bylaparoscopic debulking surgery when complete resection of all macroscopically visible disease was deemed possible » was changed by :«In brief, women diagnosed with ovarian cancer underwent a staging laparoscopy immediately followed by laparotomydebulking surgery when complete resection of all macroscopically visible disease was deemed possible. »
- I wonder why the probability of complete resection was much lower in patients with stage less than 2 a (page 6 table 4)
The authors agree with this comment. It was a mistake: the stage > IIA was correlated with less complete resection. This mistake was corrected.
- These are a number of biases in this study as the Authors suggest. For these reason I don't believe it adds any significant contribution to the literature
Authors know there is some biases in present study, but as some others publications dealing with elderly and oncology. Present study added significant contribution because very few publications try to add frailty assessment to study elderly patient and its impact in therapeutic management of patient with ovarian cancer.
Reviewer 3 Report
- Figure 1: legend of x and y axis still in French
- The use of terms group 1,2,3 and 4 makes the results difficult to interpret. Improved wording is mandatory.
- Please add an evaluation of the recurrence-free survival in the different patient groups with TR=0 and TR>0.
- Do you have data on surgical complexity score avaliable? if so this would be interesting.
- Discussion of the EWOC-1 data is missing
Author Response
The authors want to thank the reviewer for their highlights and very clever comments. Please find below answer to comments.
- Figure 1: legend of x and y axis still in French
The authors changed the legend like following.
- The use of terms group 1,2,3 and 4 makes the results difficult to interpret. Improved wording is mandatory.
The authors agree with this comment and improved wording as requested.
- Please add an evaluation of the recurrence-free survival in the different patient groups with TR=0 and TR>0.
This data could be interesting, but there are a lot of tables and figures in present work and these data could be approached with showed survival curves in the four groups. To add new figures or tables could be too much from our point of view.
- Do you have data on surgical complexity score avaliable? if so this would be interesting.
Yes, surgical complexity was shown in the table 2.
- Discussion of the EWOC-1 data is missing
The authors agree with the comment and added the EWOC-1 trial in the discussion section: “A recent French study (EWOC-1 trial) showed that for vulnerable elderly patients with ovarian cancer, carboplatin monotherapy or weekly carboplatin plus paclitaxel are often proposed as an alternative to carboplatin plus paclitaxel given every 3 weeks but they showed that compared to 3-weekly and weekly carboplatin plus paclitaxel regimens, carboplatin single agent was reported to be less active with significant worse survival outcome in vulnerable elderly patients, in vulnerable patients carboplatine paclitaxel combination remains a standard.”
Round 2
Reviewer 1 Report
The authors have addressed most of the issues raised by the reveiwer. However, there arre still some points that need to solved.
- The authors still show in table 1 that the mCCI ranges from 3-7 in the “young” group and 3-9 in the “old” group, so why is it than possible to make a cut-off <3 and >=3? The author s missed this part in their answer. Please address this.
- The authors have now added the definition of weight loss : "Weight loss was defined by a loss of weight during the year before the diagnostic of the ovarian cancer.” Though, this is still very vaguely described, as this can also be 0.1 kg over a year. Should it be more than xx kg? Please clarify
- Although, layout is improve. It still needs attention
Author Response
- Present manuscript underwent extensive English editing, performed by Felicity Neilson, from Matrix Consultant company.
- The authors still show in table 1 that the mCCI ranges from 3-7 in the “young” group and 3-9 in the “old” group, so why is it than possible to make a cut-off <3 and >=3? The author s missed this part in their answer. Please address this.
The cut-off from the mCCI score is ≤3 and > 3, but not <3 and ≥3. In the group with a mCCI score ≤3 all women are a score at 3.Woman with a mCCI score at 3 is younger than 80y without any comorbidities, she can belong to the “young” or to the “old” group. Woman with a mCCI score >3 are either young with at least one comorbidity or women older than 80 that’s why this cut-off is possible.
- The authors have now added the definition of weight loss : "Weight loss was defined by a loss of weight during the year before the diagnostic of the ovarian cancer.” Though, this is still very vaguely described, as this can also be 0.1 kg over a year. Should it be more than xx kg? Please clarify
The authors agree with this comment. They detailed this by adding the sentence in the material and methods section 2.3 data collection, ligne number 18 : Weight loss was defined by a loss of weight during the year before the diagnostic of the ovarian cancer, “i.e. loss of at least one kilogram”.
- Although, layout is improve. It still needs attention.
Layout is performed by editorial team of JCM. Authors try also to do their best.
Reviewer 2 Report
The authors report a retrospective study on ovarian cancer in the elderly population. The sample size is adequate and statistics are accurate.
Author Response
- The authors report a retrospective study on ovarian cancer in the elderly population. The sample size is adequate and statistics are accurate.
Thanks very much.
- Present manuscript was checked by a native English speaking, named Felicity Neilson who work in professional English editing service, called Matrix Consultant.
Reviewer 3 Report
Paragraph on EWOC-1 was added without a reference to the ASCO abstract.
Figure 1 was not visible to me in the pdf.
It is unclear to me why you added table 1 and 4 in the text and kept the other tables in the back.
Author Response
- Present manuscript was checked by a native English, named Felicity Neilson who work in a professional English editing service, called Matrix Consultant.
- Paragraph on EWOC-1 was added without a reference to the ASCO abstract.
The reference to the ASCO abstract was added: reference 31: Falandry C, Savoye AM, Stefani L, Tinquaut F, Lorusso D, Herrstedt J, et al. EWOC-1: A randomized trial to evaluate the feasability of three different first-line chemotherapy regimens for vulnerable elderly women with ovarian cancer (OC): A GCIG-ENGOT-GINECO study. J Clin Oncol. 2019;37(15 suppl.):5508-5508.
- Figure 1 was not visible to me in the pdf.
Authors do not know why. The figure 1 is visible in current word document.
- It is unclear to me why you added table 1 and 4 in the text and kept the other tables in the back.
Authors agree. Tables 1 and 4 were added in the back with other tables.
Round 3
Reviewer 3 Report
Accept